# Uncovering the Redundancy in Graph Self-supervised Learning Models

**Zhibiao Wang[1], Xiao Wang[1]\*, Haoyue Deng[1], Nian Liu[2], Shirui Pan[3], Chunming Hu[1]**

[1] Beihang University, China
[2] Beijing University of Posts and Telecommunications, China
[3] Griffith University, Australia

`{wzb2321, xiao_wang, haoyue_deng, hucm}@buaa.edu.cn,`
`nianliu@bupt.edu.cn, s.pan@griffith.edu.au`

## Abstract

Graph self-supervised learning, as a powerful pre-training paradigm for Graph Neural Networks (GNNs) without labels, has received considerable attention. We have witnessed the success of graph self-supervised learning on pre-training the parameters of GNNs, leading many not to doubt that whether the learned GNNs parameters are all useful. In this paper, by presenting the experimental evidence and analysis, we surprisingly discover that the graph self-supervised learning models are highly redundant at both of neuron and layer levels, e.g., even randomly removing $51.6\%$ of parameters, the performance of graph self-supervised learning models still retains at least $96.2\%$. This discovery implies that the parameters of graph self-supervised models can be largely reduced, making simultaneously fine-tuning both graph self-supervised learning models and prediction layers more feasible. Therefore, we further design a novel graph pre-training and fine-tuning paradigm called SLImming DE-correlation Fine-tuning (SLIDE[2]). The effectiveness of SLIDE is verified through extensive experiments on various benchmarks, and the performance can be even improved with fewer parameters of models in most cases. For example, in comparison with full fine-tuning GraphMAE on Amazon-Computers dataset, even randomly reducing $40\%$ of parameters, we can still achieve the improvement of $0.24\%$ and $0.27\%$ for Micro-F1 and Macro-F1 scores respectively.

## 1 Introduction

Graph self-supervised learning, aiming at learning the parameters of Graph Neural Networks (GNNs) without labels, has been a popular graph pre-training paradigm [1–3]. Usually, graph self-supervised learning is naturally divided into two learning methods, i.e., graph contrastive learning [4, 5] and graph generative self-supervised learning [6, 7]. After pre-training the parameters of self-supervised GNNs, graph self-supervised learning achieves state-of-the-art performance on a variety of tasks by fine-tuning an additional prediction layer [8, 7].

Various researches have attempted to improve and understand graph self-supervised learning from different perspectives. For example, the graph augmentation techniques [9, 10], graph spectrum feature of graph self-supervised learning [11, 12], and others [7, 13]. Despite their remarkable achievements, little efforts have been made to understand the behavior of the learned model parameters by graph self-supervised learning. Here, we ask: Are the model parameters all always useful? Or

---

\*Corresponding authors.
[2]Code available at https://github.com/zhlgg/SLIDE

what part of model parameters is useful? Whether there is the model redundancy in self-supervised GNNs? Understanding the model property in graph self-supervised learning, particularly the model redundancy, can provide valuable guidelines and insights for the development of advanced graph self-supervised learning.

For this purpose, we conduct experiments and surprisingly find out that the graph self-supervised learning models are actually highly redundant. We start with the experiments (Section 2) to investigate the downstream performance of different graph self-supervised learning models with randomly removed parameters at both neuron and layer levels. Our findings indicate that graph self-supervised learning models with even half of the neurons randomly removed have virtually no impact on the performance of the node classification task. Moreover, by closely examining the learned representations, we find out that the representations in each layer with a substantial number of neurons removed are quite similar to the representations obtained by the full set of neurons. Meanwhile, there is also a strong similarity between the representations of each layer and its adjacent layer. All results demonstrate that graph self-supervised models exhibit high model redundancy at both neuron level and layer levels.

The above findings hold great potential to improve current graph self-supervised learning models. On the one hand, it may provide valuable guideline for the pruning or the sparsity of GNNs [14, 15]. On the other hand, the phenomenon of model redundancy provides a new opportunity for designing a full fine-tuning graph self-supervised learning model. Previously, considering that the number of parameters to be fine-tuned is excessive when we directly fine-tune both GNNs and prediction layers, we usually only have to fine-tune the attached prediction layer, i.e., linear probing [3, 5, 7]. Here, if the parameters can be greatly reduced, we can simultaneously fine-tune both graph self-supervised learning models and prediction layers. Besides, the findings imply that de-correlating the learned representations is also necessary. Therefore, we propose a novel pre-training and fine-tuning paradigm called **SLIDE** by obtaining **SLI**m GNNs from the self-supervised GNNs and using the **DE**-correlation strategy to reduce the correlation between features during the fine-tuning phase. Extensive experiments on various benchmark datasets validate the effectiveness of SLIDE.

In summary, our contributions are three-fold:

- To the best of our knowledge, we are the first to uncover that graph self-supervised models exhibit high model redundancy at both neuron and layer levels. The model redundancy allows for the removal of a majority of model parameters with almost no impact on the performance of the downstream task. This discovery can significantly enhance model efficiency while maintaining acceptable task performance.

- This discovery provides two key guidelines for the subsequent graph pre-training and fine-tuning framework: one is that the parameters can be reduced, and the other is the representations should be de-correlated. These motivate us to propose a novel method, SLIDE, to achieve a pre-training and fine-tuning paradigm with fewer parameters and better performance on the downstream task.

- Comprehensive experiments demonstrate that our SLIDE outperforms baselines across multiple benchmark datasets. For example, using the Amazon-Computers dataset [16] with GRACE [1], compared to full fine-tuning, we achieve improvements of $0.37\%$ and $0.16\%$ for Macro-F1 and Micro-F1 scores respectively with a random reduction of $30\%$ of parameters.

## 2 The Model Redundancy in Graph Self-supervised Learning

In this section, we investigate the model redundancy in two representative graph self-supervised learning models (GraphMAE [3] and GRACE [1]) on node classification tasks. Specifically, we pre-train these graph self-supervised models on Cora, Citeseer, Pubmed [17], Amazon-Photo, Amazon-Computers [16], Ogbn-arxiv [18], and then we evaluate the performance again after removing neurons in various ways. If the performance gap between the original GNNs and the slim GNNs with fewer neurons is small, it indicates great model redundancy in self-supervised GNNs. In other words, even with reduced neurons, the performance remains largely unaffected, suggesting that many neurons in self-supervised GNNs are dispensable. Notably, model redundancy is not limited to node classification tasks. We conduct experiments in Appendix A demonstrating that model redundancy also persists in link prediction and graph classification tasks.

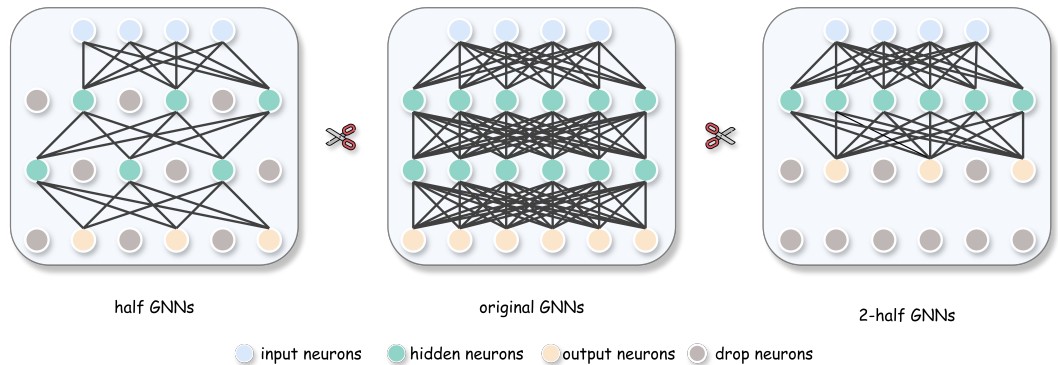

half GNNs  original GNNs  2-half GNNs

● input neurons  ● hidden neurons  ● output neurons  ● drop neurons

Figure 1: Neuron dropout. To initialize a smaller variant of the self-supervised pre-trained GNNs, we select parameters from self-supervised GNNs in different ways. **From left to right:** randomly reduce the number of neurons in each layer proportionally, the original GNNs, retain only the first two layers while randomly reducing the number of neurons in the second layer proportionally.

**Neuron removal** We explore two different ways to reduce the number of neurons in self-supervised GNNs, i.e., the original GNNs, which are illustrated in Figure 1. Specifically, we consider two approaches: (1) neuron level: randomly retaining the number of neurons by 50% and 25% in each layer named "half" and "quarter", and (2) layer level: retaining only the first two layers if the layer number is greater than two and randomly retaining the number of neurons by 100%, 50% and 25% in the second layer named "2-original", "2-half" and "2-quarter". In this way, we can obtain five types of slim GNNs: "half GNNs", "quarter GNNs", "2-original GNNs", "2-half GNNs" and "2-quarter GNNs".

Table 1: The performance of different neuron removal methods on six datasets with GraphMAE.

| Dataset | Metric | Original | Half | Quarter | 2-Original | 2-Half | 2-Quarter |
|---|---|---|---|---|---|---|---|
| Cora | F1-Mi | 83.92 | 80.36$_{\downarrow 3.56}$ | 74.88$_{\downarrow 9.04}$ | - | 82.84$_{\downarrow 1.08}$ | 81.14$_{\downarrow 2.78}$ |
| | F1-Ma | 83.01 | 79.85$_{\downarrow 3.16}$ | 73.94$_{\downarrow 9.07}$ | - | 82.06$_{\downarrow 0.95}$ | 80.31$_{\downarrow 2.70}$ |
| | Change-Param | - | $\downarrow 56.56$ | $\downarrow 79.92$ | - | $\downarrow 13.20$ | $\downarrow 19.80$ |
| CiteSeer | F1-Mi | 73.26 | 72.10$_{\downarrow 1.16}$ | 69.34$_{\downarrow 3.92}$ | - | 73.18$_{\downarrow 0.08}$ | 72.28$_{\downarrow 0.98}$ |
| | F1-Ma | 67.71 | 64.80$_{\downarrow 2.91}$ | 62.18$_{\downarrow 5.53}$ | - | 66.86$_{\downarrow 0.85}$ | 66.32$_{\downarrow 1.39}$ |
| | Change-Param | - | $\downarrow 53.03$ | $\downarrow 77.27$ | - | $\downarrow 6.10$ | $\downarrow 9.15$ |
| PubMed | F1-Mi | 80.62 | 77.94$_{\downarrow 2.68}$ | 74.86$_{\downarrow 5.76}$ | - | 79.74$_{\downarrow 0.88}$ | 77.62$_{\downarrow 3.00}$ |
| | F1-Ma | 79.97 | 77.40$_{\downarrow 2.57}$ | 74.31$_{\downarrow 5.66}$ | - | 79.06$_{\downarrow 0.91}$ | 77.17$_{\downarrow 2.80}$ |
| | Change-Param | - | $\downarrow 66.73$ | $\downarrow 87.55$ | - | $\downarrow 33.56$ | $\downarrow 50.34$ |
| Photo | F1-Mi | 93.11 | 92.75$_{\downarrow 0.36}$ | 92.26$_{\downarrow 0.85}$ | - | 92.94$_{\downarrow 0.17}$ | 92.95$_{\downarrow 0.16}$ |
| | F1-Ma | 91.91 | 91.42$_{\downarrow 0.49}$ | 90.81$_{\downarrow 1.10}$ | - | 91.65$_{\downarrow 0.26}$ | 91.63$_{\downarrow 0.28}$ |
| | Change-Param | - | $\downarrow 64.39$ | $\downarrow 85.48$ | - | $\downarrow 28.92$ | $\downarrow 43.38$ |
| Computers | F1-Mi | 90.44 | 89.49$_{\downarrow 0.95}$ | 87.87$_{\downarrow 2.57}$ | - | 90.14$_{\downarrow 0.30}$ | 89.75$_{\downarrow 0.69}$ |
| | F1-Ma | 89.24 | 88.30$_{\downarrow 0.94}$ | 86.25$_{\downarrow 2.99}$ | - | 88.92$_{\downarrow 0.32}$ | 88.55$_{\downarrow 0.69}$ |
| | Change-Param | - | $\downarrow 64.21$ | $\downarrow 85.66$ | - | $\downarrow 28.57$ | $\downarrow 42.85$ |
| arXiv | F1-Mi | 71.90 | 70.83$_{\downarrow 1.07}$ | 69.65$_{\downarrow 2.25}$ | 71.47$_{\downarrow 0.43}$ | 70.64$_{\downarrow 1.26}$ | 69.90$_{\downarrow 2.00}$ |
| | F1-Ma | 51.14 | 49.16$_{\downarrow 1.98}$ | 46.77$_{\downarrow 4.37}$ | 50.81$_{\downarrow 0.33}$ | 49.82$_{\downarrow 1.32}$ | 48.50$_{\downarrow 2.64}$ |
| | Change-Param | - | $\downarrow 73.37$ | $\downarrow 92.53$ | $\downarrow 46.96$ | $\downarrow 70.45$ | $\downarrow 82.19$ |

Table 2: The performance of different neuron removal methods on five datasets with GRACE. Ogbn-arxiv is not included because it is "out of memory" when Ogbn-arxiv is pre-trained with GRACE.

| Dataset | Metric | Original | Half | Quarter | 2-Half | 2-Quarter |
|---|---|---|---|---|---|---|
| Cora | F1-Mi | 82.30 | 80.70$_{\downarrow 1.60}$ | 77.23$_{\downarrow 5.07}$ | 81.83$_{\downarrow 0.47}$ | 79.43$_{\downarrow 2.87}$ |
| | F1-Ma | 81.12 | 79.35$_{\downarrow 1.77}$ | 75.11$_{\downarrow 6.01}$ | 80.66$_{\downarrow 0.46}$ | 78.12$_{\downarrow 3.00}$ |
| | Change-Param | - | $\downarrow 52.05$ | $\downarrow 76.54$ | $\downarrow 4.11$ | $\downarrow 6.19$ |
| CiteSeer | F1-Mi | 69.49 | 69.47$_{\downarrow 0.02}$ | 68.01$_{\downarrow 0.08}$ | 69.75$_{\uparrow 0.26}$ | 69.27$_{\downarrow 0.22}$ |
| | F1-Ma | 61.77 | 61.68$_{\downarrow 0.09}$ | 61.69$_{\downarrow 0.08}$ | 62.41$_{\uparrow 0.64}$ | 62.13$_{\downarrow 0.36}$ |
| | Change-Param | - | $\downarrow 51.62$ | $\downarrow 76.21$ | $\downarrow 3.24$ | $\downarrow 4.86$ |
| PubMed | F1-Mi | 81.14 | 79.47$_{\downarrow 1.67}$ | 76.29$_{\downarrow 4.85}$ | 81.06$_{\downarrow 0.08}$ | 80.42$_{\downarrow 0.72}$ |
| | F1-Ma | 81.05 | 79.52$_{\downarrow 1.53}$ | 75.70$_{\downarrow 5.35}$ | 80.98$_{\downarrow 0.07}$ | 80.48$_{\downarrow 0.57}$ |
| | Change-Param | - | $\downarrow 58.45$ | $\downarrow 81.34$ | $\downarrow 16.93$ | $\downarrow 25.40$ |
| Photo | F1-Mi | 91.95 | 91.40$_{\downarrow 0.55}$ | 87.90$_{\downarrow 4.05}$ | 91.35$_{\downarrow 0.60}$ | 90.95$_{\downarrow 1.00}$ |
| | F1-Ma | 90.10 | 89.40$_{\downarrow 0.70}$ | 83.76$_{\downarrow 6.34}$ | 89.40$_{\downarrow 0.70}$ | 88.95$_{\downarrow 1.15}$ |
| | Change-Param | - | $\downarrow 60.17$ | $\downarrow 82.63$ | $\downarrow 20.36$ | $\downarrow 30.54$ |
| Computers | F1-Mi | 87.57 | 85.98$_{\downarrow 1.59}$ | 83.61$_{\downarrow 3.96}$ | 86.66$_{\downarrow 0.91}$ | 85.82$_{\downarrow 1.75}$ |
| | F1-Ma | 85.84 | 84.31$_{\downarrow 1.53}$ | 81.33$_{\downarrow 4.51}$ | 85.10$_{\downarrow 0.74}$ | 84.15$_{\downarrow 1.69}$ |
| | Change-Param | - | $\downarrow 60.00$ | $\downarrow 82.50$ | $\downarrow 20.01$ | $\downarrow 30.02$ |

**Experimental results** We first pre-train the original GNNs through GraphMAE and GRACE, then we can obtain five kinds of slim GNNs by using the neuron removal methods mentioned above. For

both the slim GNNs and the original GNNs, we individually attach an additional prediction layer which is trained while keeping the original GNNs frozen on the node classification task to evaluate the Micro-F1 (F1-Mi) and Macro-F1 (F1-Ma) scores, as shown in Table 1 and Table 2, where "-" means that there are only two layers in the original GNNs (i.e.,"2-Original" and "Original" have the same performance) and "Change-Param" means the percentage change of the number of parameters compared to "Original". Surprisingly, we have following observations:

- **In all cases, "half GNNs" retain at least 96.2% of the performance of the original GNNs while the numbers of parameters are reduced by at least 51.6%.** In most of the cases, "quarter GNNs" still retain at least 90% of the performance of the original GNNs while the number of parameters are reduced by at least 76.2%. As can be seen, "quarter GNNs" of GRACE on Computers retains 95.5% of the Micro-F1 score and 94.7% of the Macro-F1 score, while the number of parameters is reduced by 82.5%.

- **The removal of layers after the second layer has a negligible impact on the performance.** The performance of "2-original GNNs" with GraphMAE on Ogbn-arxiv dataset demonstrates that even when layers after the second layer are removed, **both the Micro-F1 score and the Macro-F1 score retain 99.4% of the performance, while the number of parameters is reduced by 47.0%**.

The observations indicate that the graph self-supervised learning models are highly redundant, both at the neuron level and at the layer level. More details about the hyperparameters and the number of parameters can be found in Appendix B.

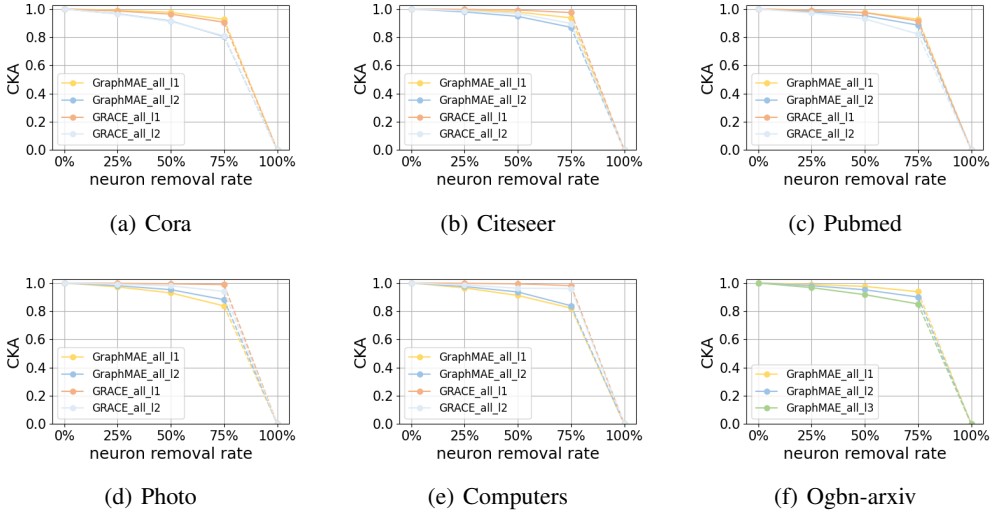

Figure 2: CKA scores between the representations of the slim GNNs and the same layer in the original GNNs with GraphMAE and GRACE on several datasets. "all" means we remove the neurons from all layers in the same proportion. "l1" means that we calculate CKA scores of the representations from the first layer, and "l2" means CKA scores from the second layer, and so on.

**Redundancy analysis on neuron level** Here, we further analyze the model redundancy by closely examining the learned representations in order to explore why the slim GNNs achieve similar performance to the original GNNs. Specifically, we get the slim GNNs by dropping the neurons of each layer in 25%, 50% and 75% to analyze the model redundancy at the neuron level. Given the learned representations of the $i$-th layer of the slim GNNs and the original GNNs, Centered Kernel Alignment (CKA) is adopted to calculate their similarity [19]. A high CKA score implies a strong similarity. As shown in Figure 2, compared to the original GNNs, the CKA scores are over $85\%$ when $50\%$ neurons are dropped. Even if the neurons are removed by $75\%$, the CKA scores are still almost over $80\%$. This indicates that removing a significant number of neurons from each layer has a minimal impact on the significance of representations of each layer and task performance consequently.

**Redundancy analysis on layer level**
As for the model redundancy at the layer level, given the learned representations of the $i$-th layer and the $(i+1)$-th layer of the original GNNs, we also adopt CKA to calculate their similarity, where the 0-th layer represents the original features of the nodes from the datasets. As shown in Figure 3, we report the CKA scores between the representations of each layer and its adjacent layer of the original GNNs for GraphMAE and GRACE. "data-layer1" means that we calculate CKA scores between the original features of the nodes from the datasets and the representations of

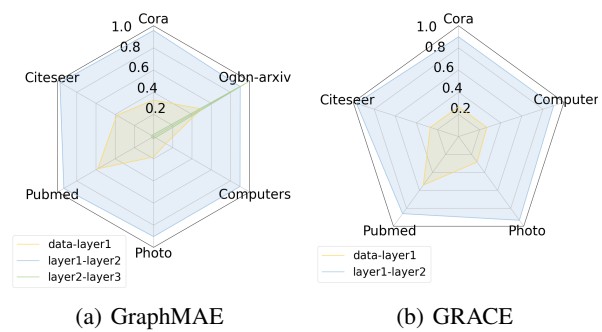

(a) GraphMAE      (b) GRACE

Figure 3: CKA scores between the representations of each layer and its adjacent layer of the original GNNs for Graph-MAE and GRACE on several datasets.

the first layer, and so on. As can be seen, CKA scores between the original features and the representations of the first layer are relatively low, while CKA scores between the representations of the layers after the first layer and their next layer are much closer to 1 in most of the cases, indicating the model redundancy at the layer level.

## 3 Our Proposed Tuning Approach: Slimming De-correlation Fine-tuning

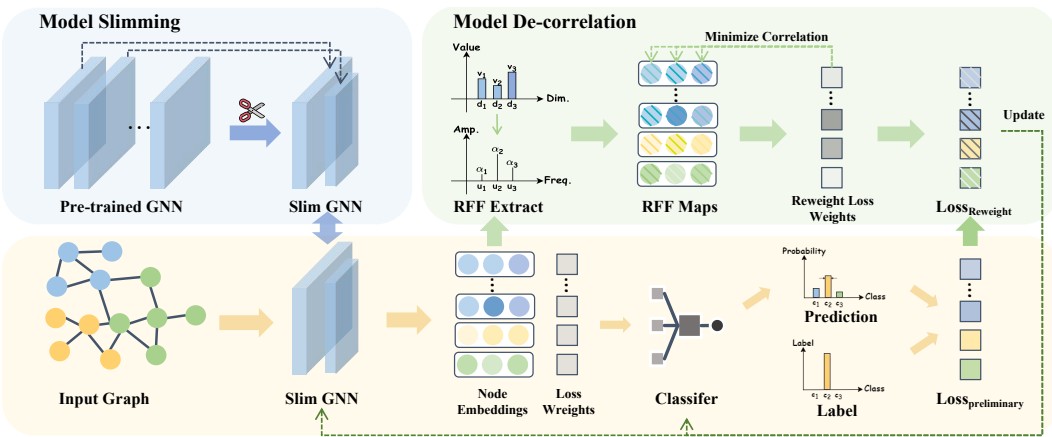

Figure 4: The overall framework of SLIDE.

In general, for the self-supervised pre-trained GNNs, we attach an additional prediction layer which is fine-tuned while keeping the GNNs frozen to conduct the downstream tasks. Ideally, if we are able to tune the parameters of both the GNNs and the linear layer, it is possible to achieve the best performance. However, the number of parameters to be fine-tuned is excessive. Here, since we identify the model redundancy in the self-supervised pre-trained GNNs, this motivates us that we actually only need to tune the additional classifier and the slim GNNs, so as to obtain a better trade-off between the model performance and the number of fine-tunable parameters. Therefore, we propose a novel pre-training and fine-tuning paradigm called **SLI**mming **DE**-correlation Fine-tuning (SLIDE), as shown in Figure 4. Specifically, firstly it reduces model redundancy in self-supervised pre-trained GNNs by randomly removing redundant neurons to obtain the slim GNNs. Then, we input the graph data into the slim GNNs to obtain the embeddings of the nodes, combined with an additional prediction layer, we can predict the label of each node. Meanwhile, we design another model de-correlation module based the squared Frobenius norm [20, 21] (an analogue corresponding to the Hilbert-Schmidt Independence Criterion, i.e., HSIC [22] in Euclidean space). The module learns the de-correlation weights for the classification loss, so as to reduce the redundancy among embeddings and make the embeddings more informative.

Specifically, let $G = (A, V, X)$ denote the input graph, where $V$ is the node set, $N = |V|$ is the number of nodes, $N_{tr}$ is the number of nodes in the training set and $d$ is the dimension of the node features, $A \in \{0,1\}^{N \times N}$ is the adjacency matrix, and $X \in R^{N \times d}$ is the input node feature matrix. For model slimming, we first pre-train the original GNNs through the existing pre-training frameworks, e.g., the graph contrastive learning [1, 4, 5] or graph generative self-supervised learning [3, 7, 2]. Then the slim GNNs $f_S$ can be obtained by randomly reducing both the neurons and the layers. Furthermore, given $G$, we can get the node embeddings $H \in R^{N \times d_h}$ as $H = f_S(A, X)$, where $d_h$ is the dimension of the node embeddings.

Motivated by Section 2 that the neurons, as well as the learned embeddings, in self-supervised GNNs are highly redundant, we aim to de-correlate the learned embeddings $H$ in the fine-tuning phase, making models with fewer parameters more informative. In particular, in inspiration of the de-correlation methods [21, 23], given the i-th dimension and j-th dimension of the node embeddings $\mathbf{H}_{*,i}$ and $\mathbf{H}_{*,j}$, we obtain Random Fourier Features (RFF) [24, 25] as

$$
\begin{aligned}
u(\mathbf{H}_{*,i}) &:= (u_1(\mathbf{H}_{*,i}), u_2(\mathbf{H}_{*,i}), \ldots, u_{N_{RFF}}(\mathbf{H}_{*,i})), \\
v(\mathbf{H}_{*,j}) &:= (v_1(\mathbf{H}_{*,j}), v_2(\mathbf{H}_{*,j}), \ldots, v_{N_{RFF}}(\mathbf{H}_{*,j})),
\end{aligned}
\tag{1}
$$

where $N_{RFF}$ is the number of functions in the random fourier space, $u_q$ and $v_q$ denote the functions from the space of Random Fourier Features. Then, we elaborate on reweighting of the weights, which encourages the independence of the node embeddings. Define the weights of the nodes in the training set as $W = \{w_n\}_{n=1}^{N_{tr}}$, where $w_n$ is the learnable weight for the $n$-th node of the training set in $G$. Consequently, the reweighted partial cross-covariance matrix can be calculated as:

$$
\begin{aligned}
\widehat{C}^{\mathbf{W}}_{\mathbf{H}_{*,i}, \mathbf{H}_{*,j}} = \frac{1}{N_{tr}-1} \sum_{n=1}^{N_{tr}} &\left[ \left( w_n u(\mathbf{H}_{n,i}) - \frac{1}{N_{tr}} \sum_{m=1}^{N_{tr}} w_m u(\mathbf{H}_{m,i}) \right)^{\top} \right. \\
&\left. \cdot \left( w_n v(\mathbf{H}_{n,j}) - \frac{1}{N_{tr}} \sum_{m=1}^{N_{tr}} w_m v(\mathbf{H}_{m,j}) \right) \right].
\end{aligned}
\tag{2}
$$

The learnable weights $W$ participate in the process of optimization to eliminate as much as possible the correlation between the dimensions of the node embeddings by minimizing the partial cross-covariance matrix in Eq. 2. Specifically, for the process of optimization, given the labels of the nodes $\mathbf{Y}_n \in R^{N_{tr}}$, we iteratively optimize the weights of the nodes $W$, the slim GNNs $f_S$, and the additional prediction layer $R$:

$$
f_S^*, R^* = \mathrm{argmin}_{f_S, R} \sum_{n=1}^{N_{tr}} w_n \ell \left( R \circ f_S \left( X_n \right), \mathbf{Y}_n \right),
\tag{3}
$$

$$
\mathbf{W}^* = \mathrm{argmin}_{\mathbf{W}} \sum_{1 \leq i < j \leq d_h} \| \widehat{C}^{\mathbf{W}}_{\mathbf{H}_{*,i}, \mathbf{H}_{*,j}} \|_{\mathrm{F}}^2,
\tag{4}
$$

where $\ell$ denotes the cross-entropy loss for the node classification task. The optimization of the weights $W$ encourages the slim GNNs $f_S$ to generate the node embeddings $H$, and eliminates the correlations between embeddings. The optimization of the slim GNNs $f_S$ and the additional classifier $R$ will lead to good performance on the node classification task.

To put it in a nutshell, SLIDE is a general pre-training and fine-tuning framework in graph, which balances the number of parameters and the performance of self-supervised pre-trained GNNs. Therefore, SLIDE can be implemented using different ideas of reducing parameters and de-correlation methods. We use the methods mentioned above as examples and conduct some experiments to demonstrate the feasibility of SLIDE.

## 4 Experiments

**Datasets.** For a comprehensive comparison, we use six real-world datasets to evaluate the performance of node classification (i.e., Cora, Citeseer, Pubmed, Amazon-Photo, Amazon-Computers and Ogbn-arxiv). More details about the datasets are in Appendix C.1.

**Baselines.** Our proposed SLIDE is a general paradigm which removes neurons from the self-supervised pre-trained GNNs randomly and introduces model de-correlation methods during the fine-tuning phase. We choose three representative graph pre-training frameworks for evaluation in our SLIDE: generative graph self-supervised learning (GraphMAE [3] and MaskGAE [7]), and graph contrastive learning (GRACE [1]). For each framework, we choose two classical fine-tuning methods as baselines: linear probing, and full fine-tuning. Notably, our proposed SLIDE is orthogonal to other fine-tuning methods. We provide additional experiments with SLIDE in Appendix C.2 as an example.

**Experimental setup.** For GraphMAE and MaskGAE, we use the implementations of their official codes [3, 7]. As for GRACE, we use the implementation of training an additional prediction layer for the node classification task, instead of using a LIBSVM classifier [26], in order to facilitate the comparison of the model's performance. For our proposed SLIDE, we use "2-half GNNs" as the slim GNNs for all datasets and pre-training frameworks. In all tables and datasets, we report averaged results along with the standard deviation computed over 5 different runs. All experiments are conducted on Linux servers equipped with NVIDIA RTX A5000 GPUs (22729 MB). We refer readers of interest to Appendix C.3 for more details on the experiments.

Table 3: Node classification accuracy ($\%\pm\sigma$) on six benchmark datasets with GraphMAE.

| Baselines | Metrics | Cora | Citeseer | Pubmed | Photo | Computers | Ogbn-arxiv |
|---|---|---|---|---|---|---|---|
| LP | F1-Mi | $83.96_{\pm0.12}\downarrow 0.32$ | $73.26_{\pm0.24}\downarrow 0.48$ | $80.62_{\pm0.17}\downarrow 0.10$ | $93.11_{\pm0.23}\downarrow 0.56$ | $90.44_{\pm0.10}\downarrow 0.41$ | $71.90_{\pm0.09}\downarrow 0.37$ |
| | F1-Ma | $83.01_{\pm0.11}\downarrow 0.28$ | $67.71_{\pm0.86}\uparrow 1.87$ | $79.97_{\pm0.15}\downarrow 0.15$ | $91.91_{\pm0.24}\downarrow 0.79$ | $89.24_{\pm0.10}\downarrow 0.64$ | $51.14_{\pm0.23}\downarrow 1.89$ |
| FT | F1-Mi | $84.10_{\pm0.30}\downarrow 0.18$ | $73.62_{\pm0.49}\downarrow 0.12$ | $80.68_{\pm0.53}\downarrow 0.04$ | $93.61_{\pm0.12}\downarrow 0.06$ | $90.61_{\pm0.33}\downarrow 0.24$ | OOM |
| | F1-Ma | $83.08_{\pm0.34}\downarrow 0.21$ | $65.21_{\pm1.12}\downarrow 0.66$ | $80.05_{\pm0.43}\downarrow 0.06$ | $92.55_{\pm0.24}\downarrow 0.15$ | $89.61_{\pm0.39}\downarrow 0.27$ | OOM |
| SLIDE | F1-Mi | $84.28_{\pm0.18}$ | $73.74_{\pm0.65}$ | $80.72_{\pm0.75}$ | $93.67_{\pm0.25}$ | $90.85_{\pm0.34}$ | $72.27_{\pm0.13}$ |
| | F1-Ma | $83.29_{\pm0.23}$ | $65.87_{\pm1.48}$ | $80.11_{\pm0.69}$ | $92.70_{\pm0.35}$ | $89.88_{\pm0.36}$ | $53.03_{\pm0.35}$ |

Table 4: Node classification accuracy ($\%\pm\sigma$) on five benchmark datasets with GRACE.

| Baselines | Metrics | Cora | Citeseer | Pubmed | Photo | Computers |
|---|---|---|---|---|---|---|
| LP | F1-Mi | $82.30_{\pm0.04}\downarrow 0.50$ | $69.49_{\pm0.12}\downarrow 2.09$ | $81.14_{\pm0.06}\downarrow 0.48$ | $91.95_{\pm0.01}\downarrow 0.98$ | $87.57_{\pm0.01}\downarrow 1.25$ |
| | F1-Ma | $81.12_{\pm0.05}\downarrow 0.05$ | $61.77_{\pm1.12}\downarrow 1.86$ | $81.05_{\pm0.06}\downarrow 0.26$ | $90.10_{\pm0.01}\downarrow 1.55$ | $85.84_{\pm0.01}\downarrow 0.89$ |
| FT | F1-Mi | $82.66_{\pm0.24}\downarrow 0.14$ | $70.38_{\pm0.58}\downarrow 1.20$ | $81.44_{\pm0.22}\downarrow 0.18$ | $92.86_{\pm0.07}\downarrow 0.07$ | $88.66_{\pm0.25}\downarrow 0.16$ |
| | F1-Ma | $81.01_{\pm0.28}\downarrow 0.16$ | $61.73_{\pm0.43}\downarrow 1.90$ | $81.11_{\pm0.22}\downarrow 0.20$ | $91.58_{\pm0.08}\downarrow 0.07$ | $86.36_{\pm0.28}\downarrow 0.37$ |
| SLIDE | F1-Mi | $82.80_{\pm0.14}$ | $71.58_{\pm0.58}$ | $81.62_{\pm0.25}$ | $92.93_{\pm0.06}$ | $88.82_{\pm0.18}$ |
| | F1-Ma | $81.17_{\pm0.13}$ | $63.63_{\pm1.03}$ | $81.31_{\pm0.25}$ | $91.65_{\pm0.09}$ | $86.73_{\pm0.34}$ |

Table 5: Node classification accuracy ($\%\pm\sigma$) on six benchmark datasets with MaskGAE.

| Baselines | Metrics | Cora | Citeseer | Pubmed | Photo | Computers | Ogbn-arxiv |
|---|---|---|---|---|---|---|---|
| LP | F1-Mi | $83.08_{\pm0.28}\downarrow 0.64$ | $72.66_{\pm0.27}\downarrow 0.74$ | $83.46_{\pm0.69}\downarrow 0.02$ | $92.94_{\pm0.11}\downarrow 0.15$ | $89.57_{\pm0.04}\downarrow 0.58$ | $71.28_{\pm0.14}\downarrow 0.12$ |
| | F1-Ma | $81.82_{\pm0.29}\downarrow 0.58$ | $67.38_{\pm0.27}\downarrow 1.93$ | $82.84_{\pm0.60}\downarrow 0.20$ | $91.72_{\pm0.15}\downarrow 0.25$ | $88.15_{\pm0.11}\downarrow 0.55$ | $49.48_{\pm0.62}\downarrow 0.94$ |
| FT | F1-Mi | $83.34_{\pm0.29}\downarrow 0.38$ | $72.42_{\pm0.19}\downarrow 0.98$ | $83.34_{\pm0.30}\downarrow 0.14$ | $93.05_{\pm0.11}\downarrow 0.04$ | $90.10_{\pm0.05}\downarrow 0.05$ | $71.34_{\pm0.21}\downarrow 0.06$ |
| | F1-Ma | $82.15_{\pm0.37}\downarrow 0.25$ | $68.92_{\pm0.19}\downarrow 0.39$ | $82.79_{\pm0.33}\downarrow 0.25$ | $91.84_{\pm0.14}\downarrow 0.13$ | $88.58_{\pm0.14}\downarrow 0.12$ | $50.51_{\pm0.25}\uparrow 0.09$ |
| SLIDE | F1-Mi | $83.72_{\pm0.19}$ | $73.40_{\pm0.52}$ | $83.48_{\pm0.40}$ | $93.09_{\pm0.12}$ | $90.15_{\pm0.10}$ | $71.40_{\pm0.24}$ |
| | F1-Ma | $82.40_{\pm0.14}$ | $69.31_{\pm0.64}$ | $83.04_{\pm0.26}$ | $91.97_{\pm0.12}$ | $88.70_{\pm0.16}$ | $50.42_{\pm0.50}$ |

## 4.1 Effectiveness of SLIDE

To evaluate our proposed SLIDE more comprehensively, we use two common evaluation metrics, Macro-F1 and Micro-F1 scores, and show their difference between baselines and SLIDE. The results

are shown in Table 3 - 5, where "LP" means "linear probe", "FT" means "full fine-tune", and "OOM" means "out of memory". We have the following observations: (1) In general, SLIDE improves the performance compared to "LP" because SLIDE is able to fine-tune both "2-half GNNs" and the additional prediction layer. For example, in comparison with "LP" on Computers with these three competitive pre-training frameworks, SLIDE achieves an average improvement of $0.75\%$ and $0.69\%$ for Micro-F1 and Macro-F1 scores, respectively. (2) Although SLIDE significantly reduces the number of parameters in self-supervised GNNs, SLIDE still achieves better performance than "FT". Especially on large-scale graphs like Ogbn-arxiv with GraphMAE, SLIDE is able to fine-tune both the pre-trained GNNs and the additional prediction layer.

## 4.2 Model Analysis

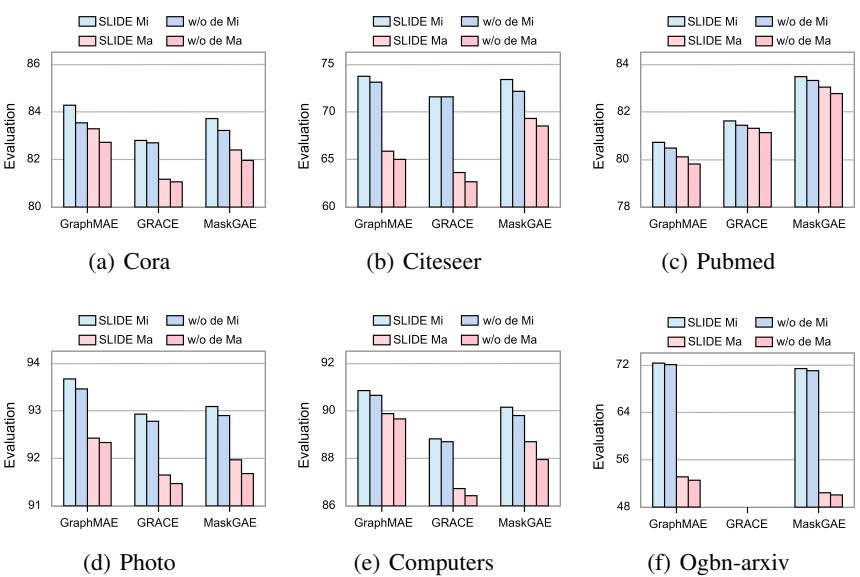

(a) Cora      (b) Citeseer      (c) Pubmed

(d) Photo      (e) Computers      (f) Ogbn-arxiv

Figure 5: Ablation studies of model de-correlation on six benchmark datasets and three pre-training frameworks. "w/o de" means that we fine-tune the slim GNNs without model de-correlation methods. "Mi" means Micro-F1 scores and "Ma" means Macro-F1 scores. The results of Ogbn-arxiv with GRACE are unseen because of "out of memory".

**Ablation study** Here, we test the performance of the slim GNNs with and without model de-correlation on the node classification task. The results are shown in Figure 5, where "w/o dec" means that the slim GNNs are directly fine-tuned without model de-correlation. We find that the slim GNNs with de-correlation perform much better than the GNNs without de-correlation, proving that correlation is still present when self-supervised GNNs are directly fine-tuned.

**Parameter analysis** In order to quantify the number of parameters of the self-supervised GNN reduced by SLIDE, taking GraphMAE and GRACE as an example, we report the number of parameters of our proposed SLIDE and "FT" for fine-tuning. As can be seen in Figure 6, we observe that the parameters of "2-half" GNNs are significantly reduced. In particular, on Ogbn-arxiv with GraphMAE, the number of parameters for fine-tuning is reduced by 70.1%. More details about the number of parameters are provided in B.1.

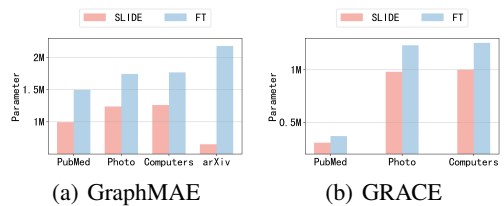

(a) GraphMAE      (b) GRACE

Figure 6: The number of parameters on several datasets with GraphMAE and GRACE.

# 5 Related Work

**Graph self-supervised learning**   Self-supervised methods on graphs can be naturally divided into contrastive and generative domains according to objective designs and model architectures [9, 27–29]. Graph Contrastive Learning (GCL) has shown its outstanding ability in unsupervised setting, and many studies have been proposed [1, 4, 5, 8]. On the other hand, Generative self-supervised learning [6, 2] aims to recover missing parts of the input data. Among them, methods which have emerged in the last two years [3, 7] have significantly enhanced the performance of generative methods, resulting in competitive performance on downstream tasks and attracting much attention. Despite the remarkable achievements of these methods, the issue of model redundancy in these self-supervised GNNs remains unexplored in the current research landscape.

**Model redundancy**   In recent years, researchers have investigated redundancy in several pre-trained model architectures for different domains. Among them, in [30], researchers dissect two pre-trained models, BERT [31] and XLNet [32], studying how much redundancy they exhibit by using feature selection to choose the subset of neurons. In [33], researchers dissected several pre-trained visual models and randomly removed neurons of the penultimate layer in proportion, proving that redundancy exists in the penultimate layer. In [34], researchers find that many layers of LLMs exhibit high similarity. By removing some of the layers of large language models (LLMs), LLMs can still maintain good performance, proving that model redundancy exists in LLMs. Graph Neural Networks (GNNs) [35–37] have been widely applied in recent years and there are some studies focusing on graph sparsification and graph lottery ticket [14, 15]. Graph sparsification approximates a graph to a sparse graph by reducing the number of edges instead of parameters. And graph lottery ticket reduces parameters in networks systematically, not randomly. However, the study of model redundancy in self-supervised GNNs remains largely unexplored.

**Pre-training and fine-tuning**   Traditional pre-training and fine-tuning paradigms mainly include "linear probe" and "full fine-tune". The former faces the challenge of insufficient performance, while the latter requires high computational cost and memory. In recent years, several Parameter-Efficient Fine-Tuning (PEFT) methods have been introduced to address these issues. Among them, Low Rank Adaptation (LoRA) [38] alters the fine-tuning phase by keeping the original model parameters frozen and introducing modifications to a separate, smaller set of parameters. These changes are then incorporated into the original parameters. On the other hand, Adapter Tuning [39] adds new modules, called adapters, between the layers of a pre-trained model. The parameters from the pre-training phase are frozen, and a smaller set of additional parameters is introduced for the new task. A common feature of these methods is the addition of a small number of additional parameters to the complete model for fine-tuning. The focus of this paper is orthogonal to these methods, as it aims to fine-tune the model under the condition of reduced parameters. In this paper, we provide a unique perspective on the pre-training and fine-tuning paradigm and contribute to the ongoing exploration of effective fine-tuning strategies.

# 6 Conclusion and Broader Impacts

In this paper, we make an exploration of model redundancy in self-supervised pre-trained GNNs. We find out that model redundancy in self-supervised GNNs exists at both neuron level and layer level, which deepens our understanding of self-supervised GNNs. We then propose a novel pre-training and fine-tuning paradigm, SLIDE, which achieves better performance with fewer number of parameters for fine-tuning. Our experiments validate the effectiveness of SLIDE.

**Limitations and broader impact**   Although we discover that the graph self-supervised learning models are highly redundant at neuron and layer levels and deepen our understanding of self-supervised GNNs, a potential limitation is that some theoretical foundations are still lacking. Our findings hold great potential to improve current graph self-supervised learning models and may provide valuable guideline for the pruning or the sparsity of GNNs. In the future, we will further understand self-supervised GNNs from the perspective of model redundancy by theoretical analysis. Beyond that, we do not expect any immediate negative impact on society.

## Acknowledgments and Disclosure of Funding

This work is supported in part by the National Natural Science Foundation of China (No. 62322203, 62172052).

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

# A More Experiences about Model Redundancy

Table 6: The performance of different neuron removal methods on three datasets with MaskGAE on link prediction tasks.

| Dataset | Metric | Original | Half | Quarter |
|---------|--------|----------|------|---------|
| Cora | AUC | 96.7 | 96.3 | 93.8 |
| | AP | 96.2 | 96.2 | 94.0 |
| | Change-Param | - | ↓ 49.9 | ↓ 74.9 |
| CiteSeer | AUC | 97.8 | 97.1 | 95.5 |
| | AP | 98.1 | 97.4 | 96.3 |
| | Change-Param | - | ↓ 50.0 | ↓ 75.0 |
| PubMed | AUC | 98.8 | 98.3 | 97.4 |
| | AP | 98.7 | 98.2 | 96.8 |
| | Change-Param | - | ↓ 49.8 | ↓ 74.8 |

Table 7: The performance of different neuron removal methods on four datasets with GraphMAE on graph classification tasks.

| Dataset | Metric | Original | Half | Quarter | 2-Original | 2-Half | 2-Quarter |
|---------|--------|----------|------|---------|------------|--------|-----------|
| MUTAG | ACC | 87.6 | 85.4 | 84.5 | 85.0 | 84.9 | 83.6 |
| | Change-Param | - | ↓ 72.1 | ↓ 91.5 | ↓ 64.6 | ↓ 70.0 | ↓ 72.7 |
| IMDB-B | ACC | 75.3 | 74.4 | 73.5 | - | 75.3 | 75.3 |
| | Change-Param | - | ↓ 71.1 | ↓ 90.8 | - | ↓ 14.2 | ↓ 21.2 |
| IMDB-M | ACC | 52.1 | 50.7 | 48.6 | 52.1 | 52.0 | 51.9 |
| | Change-Param | - | ↓ 73.2 | ↓ 92.4 | ↓ 37.4 | ↓ 46.7 | ↓ 51.4 |
| REDDIT-B | ACC | 88.2 | 85.9 | 82.5 | - | 88.0 | 87.9 |
| | Change-Param | - | ↓ 69.7 | ↓ 89.8 | - | ↓ 13.2 | ↓ 19.8 |

To demonstrate that model redundancy exists across a broader spectrum of graph learning tasks, we conduct experiments on GraphMAE, effective for graph classification, and MaskGAE which excels in link prediction. For graph classification, we conduct experiments on 4 benchmarks: MUTAG, IMDB-B, IMDB-M, REDDIT-B [40]. The results in Table 6 - 7 indicate that model redundancy exists across a wide range of tasks.

# B More Details about Model Redundancy

## B.1 GNN Parameters and Linear Parameters

Table 8: More details about paramters with different neuron removal methods for GraphMAE, where the parameters in GNN is not fine-tunable while the parameters in Linear is fine-tunable.

| Dataset | Parameters | Original | Half | Quarter | 2-Original | 2-Half | 2-Quarter |
|---------|-----------|----------|------|---------|------------|--------|-----------|
| Cora | GNN | 998,914 | 433,922 | 200,578 | - | 867,074 | 801,154 |
| | Linear | 3,591 | 1,799 | 903 | - | 1,799 | 903 |
| CiteSeer | GNN | 2,161,154 | 1,015,042 | 491,138 | - | 2,029,314 | 1,963,394 |
| | Linear | 3,078 | 1,542 | 774 | - | 1,542 | 774 |
| PubMed | GNN | 1,566,722 | 521,218 | 195,074 | - | 1,040,898 | 777,986 |
| | Linear | 3,075 | 1,539 | 771 | - | 1,539 | 771 |
| Photo | GNN | 1,821,698 | 648,706 | 258,818 | - | 1,294,850 | 1,031,426 |
| | Linear | 8,200 | 4,104 | 2,056 | - | 4,104 | 2056 |
| Computers | GNN | 1,844,226 | 659,970 | 264,450 | - | 1,317,378 | 1,053,954 |
| | Linear | 10,250 | 5,130 | 2,570 | - | 5,130 | 2,570 |
| arXiv | GNN | 2,243,587 | 597,507 | 167,683 | 1,189,890 | 663,042 | 399,618 |
| | Linear | 41,000 | 20,520 | 10,280 | 41,000 | 20,520 | 10280 |

Table 9: More details about paramters with different neuron removal methods for GRACE.

| Dataset | Parameters | Original | Half | Quarter | 2-Half | 2-Quarter |
|---------|-----------|----------|------|---------|--------|-----------|
| Cora | GNN | 400,000 | 191,808 | 93,856 | 383,552 | 375,328 |
| | Linear | 903 | 455 | 231 | 455 | 231 |
| CiteSeer | GNN | 2,027,777 | 981,121 | 482,369 | 1,962,113 | 1,929,281 |
| | Linear | 1542 | 774 | 390 | 774 | 390 |
| PubMed | GNN | 387,840 | 161,152 | 72,384 | 322,176 | 289,344 |
| | Linear | 771 | 387 | 195 | 387 | 195 |
| Photo | GNN | 1,288,705 | 513,281 | 223,873 | 1,026,305 | 895,105 |
| | Linear | 4104 | 2056 | 1032 | 2056 | 1032 |
| Computers | GNN | 1,311,232 | 524,544 | 229,504 | 1,048,832 | 917,632 |
| | Linear | 5130 | 2570 | 1290 | 2570 | 1290 |

Tables 8 and Table 9 record the number of GNN parameters and Linear parameters for different ways of removing neurons. We find out that proper removal of neurons still maintains decent performance, as we show in Section 2, which means that there is a lot of model redundancy in the original GNNs. Note that we train the same number of epochs for the different ways of reducing neurons to save model training time, and the gap in model performance will be smaller, i.e. there will still be more model redundancy, if training time is more.

## B.2   Hyper-parameters for GraphMAE and GRACE

For datasets where the original pre-training framework has been tested, we use the hyper-parameters from the official code, while for the other datasets, we obtain the hyper-parameters ourselves by testing on these datasets.

For GraphMAE, we obtain the hyper-parameters of pre-training on Amazon-Photo and Amazon-Computers by ourselves. For both datasets, linear probes are trained using Adam with a learning rate of 0.01, momentum of 0.9 and weight decay of 0.0005 while GNNs are pre-trained with a learning rate of 0.001, weight decay of 0, hidden number of 1024, head number of 4, layer number of 2, mask rate of 0.5, drop edge rate of 0.5 and epoch number of 1000.

For GRACE, we obtain the hyper-parameters of all linear probes on all datasets by ourselves because it trains a LIBSVM classifier in the official code while we obtain the the hyper-parameters of pre-training on Amazon-Photo and Amazon-Computers by ourselves. Here we list the hyper-parameters for pre-trained models and linear probes used in our experiments:

- For Cora, the linear probe is trained using Adam with a learning rate of 0.05, momentum of 0.9, epoch number of 1000 and weight decay of 0.

- For Citeseer, the linear probe is trained using Adam with a learning rate of 0.5, momentum of 0.9, epoch number of 1000 and weight decay of 0.

- For Pubmed, the linear probe is trained using Adam with a learning rate of 0.05, momentum of 0.9, epoch number of 500 and weight decay of 0.

- For Photo, the linear probe is trained using Adam with a learning rate of 0.05, momentum of 0.9, epoch number of 500 and weight decay of 0 whlie GNN is pre-trained with a learning rate of 0.001, weight decay of 0, hidden number of 512, layer number of 2, and epoch number of 200.

- For Computers, the linear probe is trained using Adam with a learning rate of 0.5, momentum of 0.9, epoch number of 500 and weight decay of 0 whlie GNN is pre-trained with a learning rate of 0.001, weight decay of 0, hidden number of 256, layer number of 2, and epoch number of 200.

Table 10: Dataset Statistics

| Datasets | # Nodes | # Edges | # Features | # Classes | Split ratio |
|----------|---------|---------|-----------|-----------|-------------|
| Cora | 2,708 | 10,556 | 1,433 | 7 | 140/500/1,000 |
| Citeseer | 3,327 | 9,104 | 3,703 | 6 | 120/500/1,000 |
| Pubmed | 19,717 | 88,648 | 500 | 3 | 60/500/1,000 |
| Photo | 7,650 | 238,162 | 745 | 8 | 10%/10%/80% |
| Computers | 13,752 | 491,722 | 767 | 10 | 10%/10%/80% |
| arXiv | 16,9343 | 2,315,598 | 128 | 40 | 90,941/29,799/48,603 |

# C Experimental Details

## C.1 Datasets and Pre-training Frameworks

Here, we give some details about datasets we choose to evaluate the performance of SLIDE. As we have mentioned in Section 4, we use several citation networks and two social networks and Ogbn-arxiv datasets. Among these, the edges in citation networks (i.e. Cora, Citeseer, and Pubmed) represent the citation relationship between two papers (undirected), the node features are the bag-of-words vector of the papers, and the labels are the fields of the papers. The nodes in social networks (i.e. Amazon-Photo and Amazon-Computers) represent the products, the edges represent whether the two products are frequently purchased together, the features represent the product reviews encoded in bag-of-words, and the labels are the predefined product categories. Ogbn-arxiv captures citation relationships between computer science papers on arxiv. Nodes denote papers, edges denote citation relationships of papers, and each paper has a dimensional feature vector obtained by averaging the embeddings of words in the title and abstract. The embeddings are obtained using Word2Vec [41]. The test is to predict 40 domains over CS.

For the implementations of three pre-training frameworks, we use their original code. The sources are listed as follows:

1. GraphMAE: https://github.com/THUDM/GraphMAE

2. GRACE: https://github.com/CRIPAC-DIG/GRACE

3. MaskGAE: https://github.com/EdisonLeeeee/MaskGAE

## C.2 Additional Experiments and Analysis of SLIDE with Fine-Tuning Methods

Table 11: Orthogonality experiment of our proposed SLIDE and traditional fine-tuning methods, using LoRA as an example.

| Dataset | Metric | Linear-probing | LoRA | Slim-LoRA | SLIDE-LoRA |
|---------|--------|----------------|------|-----------|------------|
| Cora | ACC | $83.96_{\pm 0.12}$ | $84.18_{\pm 0.34}$ | $83.62_{\pm 0.29}$ | $84.26_{\pm 0.43}$ |
| CiteSeer | ACC | $73.26_{\pm 0.24}$ | $73.27_{\pm 0.36}$ | $72.88_{\pm 0.51}$ | $73.37_{\pm 0.57}$ |
| PubMed | ACC | $80.62_{\pm 0.17}$ | $80.69_{\pm 0.61}$ | $80.36_{\pm 0.63}$ | $80.63_{\pm 0.65}$ |

Here we examine the orthogonality of SLIDE in relation to traditional fine-tuning methods. We start with the classic LoRA method [38] applied to GraphMAE, where SLIDE randomly prunes a subset of neurons to create Slim GNNs. We then introduce an additional LoRA module designed for fine-tuning. The results are shown in Table 11. Notably, "SLIDE-LoRA" can only adjust the parameters of the LoRA modules, as the Slim GNNs remain fixed. Despite this limitation, "SLIDE-LoRA" enhances performance by reducing correlations among final representations, achieving slightly better results compared to using LoRA directly on Original GNNs. This supports the efficacy of our method in improving model capabilities.

## C.3 Experimental Settings

Here, we provide more experimental settings about the experience about SLIDE. We obtain the hyper-parameters ourselves by testing on these datasets with three frameworks except linear probing (we obtain the hyper-parameters from the official code).

Here we list the hyper-parameters for full fine-tuning:

- GraphMAE:
  - Cora: The linear probe is trained using Adam with a learning rate of 0.05, momentum of 0.9 and weight decay of 1e-4 whlie GNN is tuned with a learning rate of 1e-7, weight decay of 0.
  - Citeseer: The linear probe is trained using Adam with a learning rate of 0.02, momentum of 0.9 and weight decay of 1e-1 whlie GNN is tuned with a learning rate of 1e-6, weight decay of 1e-3.
  - Pubmed: The linear probe is trained using Adam with a learning rate of 0.05, momentum of 0.9 and weight decay of 0 whlie GNN is tuned with a learning rate of 1e-6, weight decay of 0.
  - Photo: The linear probe is trained using Adam with a learning rate of 0.01, momentum of 0.9 and weight decay of 0.05 whlie GNN is tuned with a learning rate of 5e-6, weight decay of 0.
  - Computers: The linear probe is trained using Adam with a learning rate of 0.01, momentum of 0.9 and weight decay of 0.05 whlie GNN is tuned with a learning rate of 5e-5, weight decay of 0.
  - Ogbn-arxiv: The linear probe is trained using Adam with a learning rate of 0.02, momentum of 0.9 and weight decay of 0 whlie GNN is tuned with a learning rate of 5e-4, weight decay of 1e-3.

- GRACE:
  - Cora: The linear probe is trained using Adam with a learning rate of 0.02, momentum of 0.9 and weight decay of 0 whlie GNN is tuned with a learning rate of 1e-7, weight decay of 0.
  - Citeseer: The linear probe is trained using Adam with a learning rate of 0.01, momentum of 0.9 and weight decay of 0.01 whlie GNN is tuned with a learning rate of 1e-8, weight decay of 0.
  - Pubmed: The linear probe is trained using Adam with a learning rate of 0.02, momentum of 0.9 and weight decay of 0 whlie GNN is tuned with a learning rate of 1e-6, weight decay of 0.
  - Photo: The linear probe is trained using Adam with a learning rate of 0.02, momentum of 0.9 and weight decay of 0 whlie GNN is tuned with a learning rate of 5e-5, weight decay of 0.
  - Computers: The linear probe is trained using Adam with a learning rate of 0.1, momentum of 0.9 and weight decay of 0 whlie GNN is tuned with a learning rate of 5e-4, weight decay of 0.

- MaskGAE:
  - Cora: The linear probe is trained using Adam with a learning rate of 5e-3, momentum of 0.9 and weight decay of 1e-3 whlie GNN is tuned with a learning rate of 1e-4, weight decay of 1e-3.
  - Citeseer: The linear probe is trained using Adam with a learning rate of 0.01, momentum of 0.9 and weight decay of 5e-3 whlie GNN is tuned with a learning rate of 1e-4, weight decay of 1e-4.
  - Pubmed: The linear probe is trained using Adam with a learning rate of 0.015, momentum of 0.9 and weight decay of 5e-4 whlie GNN is tuned with a learning rate of 1e-4, weight decay of 0.
  - Photo: The linear probe is trained using Adam with a learning rate of 0.01, momentum of 0.9 and weight decay of 0.01 whlie GNN is tuned with a learning rate of 1e-4, weight decay of 0.

- Computers: The linear probe is trained using Adam with a learning rate of 5e-3, momentum of 0.9 and weight decay of 5e-3 whlie GNN is tuned with a learning rate of 2e-4, weight decay of 0.
- Ogbn-arxiv: The linear probe is trained using Adam with a learning rate of 0.01, momentum of 0.9 and weight decay of 0 whlie GNN is tuned with a learning rate of 1e-4, weight decay of 0.

Here we list the hyper-parameters for SLIDE:

- GraphMAE:
  - Cora: The linear probe is trained using Adam with a learning rate of 0.05, momentum of 0.9 and weight decay of 1e-4 whlie GNN is tuned with a learning rate of 1e-7, weight decay of 0.
  - Citeseer: The linear probe is trained using Adam with a learning rate of 0.02, momentum of 0.9 and weight decay of 1e-1 whlie GNN is tuned with a learning rate of 1e-6, weight decay of 1e-3.
  - Pubmed: The linear probe is trained using Adam with a learning rate of 0.05, momentum of 0.9 and weight decay of 0 whlie GNN is tuned with a learning rate of 1e-6, weight decay of 0.
  - Photo: The linear probe is trained using Adam with a learning rate of 0.01, momentum of 0.9 and weight decay of 0.05 whlie GNN is tuned with a learning rate of 5e-6, weight decay of 0.
  - Computers: The linear probe is trained using Adam with a learning rate of 0.01, momentum of 0.9 and weight decay of 0.05 whlie GNN is tuned with a learning rate of 5e-5, weight decay of 0.
  - Ogbn-arxiv: The linear probe is trained using Adam with a learning rate of 0.02, momentum of 0.9 and weight decay of 0 whlie GNN is tuned with a learning rate of 5e-4, weight decay of 1e-3.

- GRACE:
  - Cora: The linear probe is trained using Adam with a learning rate of 0.02, momentum of 0.9 and weight decay of 0 whlie GNN is tuned with a learning rate of 1e-7, weight decay of 0.
  - Citeseer: The linear probe is trained using Adam with a learning rate of 0.01, momentum of 0.9 and weight decay of 0.01 whlie GNN is tuned with a learning rate of 1e-8, weight decay of 0.
  - Pubmed: The linear probe is trained using Adam with a learning rate of 0.02, momentum of 0.9 and weight decay of 0 whlie GNN is tuned with a learning rate of 1e-6, weight decay of 0.
  - Photo: The linear probe is trained using Adam with a learning rate of 0.02, momentum of 0.9 and weight decay of 0 whlie GNN is tuned with a learning rate of 5e-5, weight decay of 0.
  - Computers: The linear probe is trained using Adam with a learning rate of 0.1, momentum of 0.9 and weight decay of 0 whlie GNN is tuned with a learning rate of 5e-4, weight decay of 0.

- MaskGAE:
  - Cora: The linear probe is trained using Adam with a learning rate of 0.05, momentum of 0.9 and weight decay of 1e-3 whlie GNN is tuned with a learning rate of 1e-4, weight decay of 5e-3.
  - Citeseer: The linear probe is trained using Adam with a learning rate of 0.01, momentum of 0.9 and weight decay of 0.01 whlie GNN is tuned with a learning rate of 5e-4, weight decay of 1e-4.
  - Pubmed: The linear probe is trained using Adam with a learning rate of 0.02, momentum of 0.9 and weight decay of 1e-3 whlie GNN is tuned with a learning rate of 1e-4, weight decay of 1e-2.
  - Photo: The linear probe is trained using Adam with a learning rate of 0.02, momentum of 0.9 and weight decay of 6e-3 whlie GNN is tuned with a learning rate of 1e-4, weight decay of 0.

- Computers: The linear probe is trained using Adam with a learning rate of 5e-3, momentum of 0.9 and weight decay of 5e-3 whlie GNN is tuned with a learning rate of 1e-4, weight decay of 0.
- Ogbn-arxiv: The linear probe is trained using Adam with a learning rate of 3e-3, momentum of 0.9 and weight decay of 0 whlie GNN is tuned with a learning rate of 2e-4, weight decay of 0.

And the hyper-parameters of linear probing is the same as the config file of these pre-training frameworks. The hyper-parameters are different sometimes because their model structures are different, the parameters used to achieve optimal performance are sometimes different. And the hyper-parameters of the slim GNNs without model de-correlation are the same with SLIDE.

