# OpenReview forum: "Uncovering the Redundancy in Graph Self-supervised Learning Models"
_NeurIPS.cc/2024/Conference — NeurIPS 2024 poster_

### Official Review · Reviewer_1uus · 2024-07-08

**Soundness:** 3
**Presentation:** 3
**Contribution:** 4
**Rating:** 7
**Confidence:** 4

**Summary:**

This paper presents new insights on graph self-supervised learning models. Namely, the parameters, as well as the learned representations, of graph self-supervised learning models are highly redundant. The paper also proposes a novel pre-training and fine-tuning paradigm, SLIDE, which achieves better performance with fewer number of parameters for fine-tuning. The experimental results are also remarkable, e.g., the improvements can be also achieved even with a random reduction of parameters.

**Strengths:**

- Deepening our understanding of graph self-supervised learning models is an important endeavor, given the popularity of these models. This paper is aligned with research lines and proposes some important discovers on redundancy in terms of both neuron and layer levels.
- The paper is overall well-organized and clearly written. I enjoy reading it.
- The experiments are clearly designed and well-executed, further demonstrating the value of this work.

Overall, this paper is a valuable contribution to the field and deserves a wide audience for its discovers and insights.

**Weaknesses:**

I think it would be valuable to explain more on the empirical results. I in particular wonder why “Although SLIDE significantly reduces the number of parameters in self-supervised GNNs, SLIDE still achieves better performance than FT".
Is there any (mathematical) definition on the model redundancy? It is better to verify that the proposed model does reduce the model redundancy.

**Questions:**

see weakness

---

> ### Author Rebuttal · Authors · 2024-08-06
>
> We sincerely thank you for all the comments and it is a great honor for us for your enjoying our paper. We have addressed all your questions below and hope they have clarified all confusion you had about our work.
>
> > [W\#1] I in particular wonder why “Although SLIDE significantly reduces the number of parameters in self-supervised GNNs, SLIDE still achieves better performance than FT".
>
> **Response:**
>
> Indeed, reducing the number of parameters can lead to a slight performance decrease in self-supervised GNNs, specifically when applying FT to Slim GNNs compared to Original GNNs. However, as highlighted in Section 2, we identify strong correlations among different dimensions of embeddings, emphasizing the necessity of introducing de-correlation techniques to enhance the informativeness of graph self-supervised learning models. Therefore, the improved performance of SLIDE despite having fewer parameters is attributed to the incorporation of decorrelation methods. This underscores the effectiveness of our approach in mitigating model redundancy and enhancing model performance. Moreover, in Section 4.2 of our paper we conduct ablation experiments on the de-correlation method, validating our findings in Section 2 and demonstrating its critical importance within SLIDE.
>
> > [W\#2] Is there any (mathematical) definition on the model redundancy? It is better to verify that the proposed model does reduce the model redundancy.
>
> **Response:** We appreciate and thank you for your insightful question.
>
> The redundancy of the model currently lacks a mathematical definition, but can be roughly measured using some alternative indicators. Typically, we evaluate the model redundancy by assessing the performance of the models and the correlations within the representations. In our paper, we highlight the beneficial performance of the SLIDE models, which suggests a reduction in model redundancy. As for correlations, we conduct layer-level and neuron-level correlation analyses on embeddings from both Original GNN and the SLIDE models, and these results are presented in the table below. To simplify the evaluation process, we analyze representations of randomly sampled nodes and we use MaskGAE as an example. At the layer level, we assess the correlations between representations of adjacent layers. At the neuron level, we calculate correlations between representations represented by different subsets of neurons within a single layer. All these results are computed using CKA scores. Furthermore, as mentioned in the Conclusion, we recognize the importance of supplementing theoretical analyses for model redundancy, which is a focus of our future work. We aim to provide a detailed mathematical definition of model redundancy in our future endeavors. Once again, thank you for your insightful questions and your appreciation of our work!
>
> **Layer Level:**
>
> |    Module     |    Metric     |  Cora  | CiteSeer | PubMed |
> | :-----------: | :-----------: | :----: | :------: | :----: |
> |               |  data-layer1  | 0.3939 |  0.3131  | 0.5141 |
> | Original GNNs | layer1-layer2 | 0.9471 |  0.9509  | 0.9405 |
> |               |    average    | 0.6705 |  0.6320  | 0.7273 |
> |               |  data-layer1  | 0.3834 |  0.2680  | 0.4881 |
> |     SLIDE     | layer1-layer2 | 0.8622 |  0.8413  | 0.8945 |
> |               |    average    | 0.6228 |  0.5546  | 0.6963 |
>
> **Neuron Level:**
>
> |    Module     | Metric  |  Cora  | CiteSeer | PubMed |
> | :-----------: | :-----: | :----: | :------: | :----: |
> |               | layer1  | 0.6957 |  0.7567  | 0.7596 |
> | Original GNNs | layer2  | 0.7566 |  0.7772  | 0.7652 |
> |               | average | 0.7262 |  0.7670  | 0.7624 |
> |               | layer1  | 0.6768 |  0.7462  | 0.7394 |
> |     SLIDE     | layer2  | 0.5641 |  0.5637  | 0.7056 |
> |               | average | 0.6204 |  0.6550  | 0.7225 |

---

### Official Review · Reviewer_oLUe · 2024-07-08

**Soundness:** 3
**Presentation:** 3
**Contribution:** 3
**Rating:** 6
**Confidence:** 4

**Summary:**

This paper studies the redundancy in graph self-supervised learning models. The authors discover that even randomly removing a number of parameters, the performance of graph self-supervised learning models is still comparable, revealing the redundancy problem. Then the authors propose to simultaneously fine-tune the graph self-supervised learning models and the prediction layers, and the effectiveness are well demonstrated on various benchmarks.

**Strengths:**

Discovery of Redundancy: The paper identifies an intriguing phenomenon of redundancy within graph self-supervised learning models. This finding provides valuable insights for future research directions, particularly in the full fine-tuning of these models. Clear Organization: The paper is well-organized and logical, featuring high-quality tables and figures that effectively support the presented data and analyses. Strong Performance: The performance of the proposed model is robust, showing significant improvements when compared to baseline models.

**Weaknesses:**

Some technique details in the paper are not clearly introduced, leaving several critical aspects insufficiently explained. For example, the rationale behind the use of Random Fourier Features (RFF) is not adequately addressed. It would be beneficial to provide more in-depth explanations on why RFF was chosen for this context and how it contributes to the model's performance. Additionally, the meaning and significance of Equation 2 are not thoroughly discussed, leaving readers uncertain about its role and impact within the overall framework. The paper also lacks an analysis of the learned weights 𝑊 which is crucial for understanding the model's learning dynamics and interpretability. Including such an analysis could offer valuable insights into how the model processes and prioritizes different features. Moreover, the inclusion of more experimental results would enhance the robustness of the findings. By providing a more comprehensive set of experiments, the paper could demonstrate the consistency and generalizability of the proposed approach across different scenarios and datasets.

**Questions:**

Please refer to the weakness above.

**Limitations:**

Yes

---

> ### Author Rebuttal · Authors · 2024-08-06
>
> We are deeply grateful for your insightful feedback and constructive suggestions. Your thorough review has significantly strengthened the quality of our manuscript.
>
> >[W\#1] About RFF
>
> **Response:** Thanks for your good question about Random Fourier Features (RFF).
>
> Deep neural networks exhibit complex dependencies between their features, where simply removing linear correlations is insufficient. A direct approach to address this challenge involves mapping features to a high-dimensional space using kernel methods. However, kernel mapping expands the original feature map to an infinite dimension, rendering it impractical to compute correlations between dimensions. Recognizing the advantageous properties of Random Fourier Features (RFF) in approximating kernel functions and measuring feature independence, SLIDE employs RFF to project original features into the function space of Random Fourier Features. Then, we eliminate the linear correlations among new features, thereby removing both linear and non-linear correlations among the original features.
>
> Without RFF, as shown in Equation 4, the regularization mechanism would degrade, focusing only on linear correlations. This highlights the critical role of RFF in enabling the model to effectively capture and model non-linear relationships. This capability significantly enhances the model's capacity to learn intricate data patterns, which is particularly advantageous in tasks such as node classification.
>
> >[W\#2] About Equation 2
>
> **Response:**
> $\hat{C}\_{H\_{\*,i},H\_{\*,j}}^W=\frac{1}{N_{tr}-1}\sum_{n=1}^{N_{tr}}[(w_nu(H_{n,i})-\frac{1}{N_{tr}}\sum_{m=1}^{N_{tr}}w_mu(H_{m,i}))^T\\ \cdot (w_nv(H_{n,j})-\frac{1}{N_{tr}}\sum_{m=1}^{N_{tr}}w_mv(H_{m,j}))]$
>
> For the meaning of Equation 2, $H_{\*, i}$ and $H_{\*, j}$ mean the i-th and j-th dimension of node embeddings. $W$ means the learnable weights of nodes in the input graph data, and $w_n$ means the weight of the n-th nodes in the training set which has $N_{tr}$ nodes. $\hat{C}\_{H\_{\*,i},H\_{\*,j}}^{W}$ is the partial cross-covariance matrix. What's more, $u$ and $v$ are the random Fourier features function space. Overall, Equation 2 aims to obtain the partial cross-covariance matrix between different dimensions of embeddings through node weighting.
> And for the significance of Equation 2, it facilitates the minimization of correlations across different dimensions of embeddings by learning the weights W associated with nodes. This approach enables the framework to effectively manage and optimize relationships among different dimensions of embeddings, thereby reducing the model redundancy and enhancing the model's performance.
>
> >[W\#3] About Weights W
>
> **Response:** Thanks for your valuable suggestions.
>
> Using MaskGAE on the Cora dataset, we analyze the learned weights W with respect to node degrees. We observe that nodes with smaller degrees tend to have larger weights. Across multiple epochs and runs, we identify the top five nodes with the highest weights for each epoch (referred to as "big answer"). The 8th node consistently appears prominently, representing about 10% of occurrences, while over half of the nodes rarely appear in the "big answer". These findings highlight important nodes in the dataset, suggesting further investigation into their significance.
>
> Our hypothesis links these observations to the graph's underlying structure. We examine the relationship between node degrees and weights by comparing the average degree of nodes in the training set with those of the top ten nodes most and least frequent in the "big answer". Nodes with degrees ≤ 4 average 125.0222 occurrences, while those ≥ 10 average 37.8723 occurrences, reinforcing our hypothesis. We sincerely thank for this insightful question, which has prompted further exploration into an intriguing aspect of our analysis.
>
> >[W\#4] About the consistency and generalizability of the proposed approach
>
> **Response:** Thank you for your valuable feedback and suggestion.
>
> In our study, we conduct experiments to investigate model redundancy in graph self-supervised learning models on the classical node classification tasks. These preliminary experiments lay the foundation for our findings. We acknowledge the importance of expanding our analysis to include more diverse graph learning tasks, and we have conducted initial experiments about model redundancy in graph self-supervised learning models on graph classification and link prediction tasks. The results are shown below, illustrating that model redundancy in graph self-supervised learning extends across a broad spectrum of graph learning tasks. This will allow us to further validate and generalize the effectiveness of our approach across different scenarios and datasets. We appreciate your insights and look forward to incorporating them into our future work.
>
> **Link prediction:**
>
> |Dataset|Metric|Original|Half|Quarter|
> |:------:|:----------:|:--------:|:--------:|:--------:|
> ||AUC| 96.66 | 96.27 | 93.80 |
> |Cora|AP|96.21|96.21|93.99|
> ||Change-Param |\-|49.94|74.90|
> ||AUC|97.84|97.12|95.52|
> |CiteSeer|AP|98.06|97.37|96.25|
> ||Change-Param |\-|49.97|74.96|
> ||AUC|98.79|98.31|97.38|
> |PubMed|AP|98.71|98.15|96.81|
> ||Change-Param|\-|49.84|74.76|
>
> **Graph classification:**
>
> |Dataset|Metric|Original|Half|Quarter|2-Original|2-Half|2-Quarter|
> |:-----------:|:----------:|:--------:|:--------:|:--------:|:--------:|:--------:|:--------:|
> |MUTAG|ACC|87.56|85.36|84.51|85.02|84.92|83.64|
> ||Change-Param|\-|72.08|91.53|64.61|69.99|72.68|
> |IMDB-BINARY|ACC|75.32|74.40|73.54|\-|75.34|75.28|
> ||Change-Param|\-|71.11|90.83|\-|14.16|21.23|
> |IMDB-MULTI|ACC|52.12|50.71|48.55|52.05|52.01|51.91|
> ||Change-Param|\-|73.23|92.42|37.38|46.73|51.41|
> |REDDIT-BINARY|ACC|88.24|85.93|82.47|\-|88.03|87.93|
> ||Change-Param|\-|69.70|89.78|\-|13.21|19.82|

---

### Official Review · Reviewer_nfbW · 2024-07-09

**Soundness:** 3
**Presentation:** 3
**Contribution:** 4
**Rating:** 7
**Confidence:** 4

**Summary:**

This paper is the first to uncover that graph self-supervised models exhibit high model redundancy at both neuron and layer levels, providing two key perspectives for graph pre-training and fine-tuning framework.
This paper proposes SLIDE to achieve a pre-training and fine-tuning paradigm with fewer parameters and better performance on the downstream task.
The authors conduct comprehensive experiments, showing superior performance.

**Strengths:**

Overall, this paper is in general of good quality that it is well organized and in general clearly written. The motivation is clear and strong.
The problem under investigation is an interesting problem and the work offers some important discovers and results for this problem.
The proposed method is simple, effective, and well-motivated with excellent performance. Some results are particularly impressive (e.g., even randomly reducing 40% of parameters, the improvement is still 0.24% and 0.27%)

**Weaknesses:**

The task evaluated in this paper is only node classification. I’m curious what about other tasks? It might be important to check the correlation between the model redundancy problem and node classification, or the problem is task-independent?
 The authors point out that the proposed model is orthogonal to other fine tuning methods. A detailed discussion or more experimental results would be a plus.

**Questions:**

See weakness section

---

> ### Author Rebuttal · Authors · 2024-08-06
>
> We extend our sincere gratitude for the thoughtful review and constructive feedback. Your thorough evaluation has undoubtedly contributed to enhancing the robustness of our findings.
>
> >[W\#1]  The task evaluated in this paper is only node classification. I’m curious what about other tasks? It might be important to check the correlation between the model redundancy problem and node classification, or the problem is task-independent?
>
> **Response:** Thanks for your question.
>
> We choose node classification as a representative task for evaluating model redundancy in graph self-supervised learning due to its classic and widely studied nature in graph learning. However, the issue of model redundancy is not limited to node classification. We further conduct similar experiments on other graph learning tasks, i.e., graph classification and link prediction, using GraphMAE and MaskGAE, respectively. The results are consistent with those observed in node classification, demonstrating that the problem of model redundancy is indeed pervasive across different graph self-supervised learning tasks.
>
> We appreciate your interest in this aspect and hope that these additional results provide a comprehensive understanding of the issue across various tasks.
>
> **Link prediction:**
>
> | Dataset  |    Metric    |  Original  |    Half    |  Quarter   |
> | :------: | :----------: | :--------: | :--------: | :--------: |
> |          |     AUC      | 96.66±0.08 | 96.27±0.06 | 93.80±0.20 |
> |   Cora   |      AP      | 96.21±0.17 | 96.21±0.06 | 93.99±0.22 |
> |          | Change-Param |     \-     |   49.94    |   74.90    |
> |          |     AUC      | 97.84±0.12 | 97.12±0.11 | 95.52±0.06 |
> | CiteSeer |      AP      | 98.06±0.10 | 97.37±0.07 | 96.25±0.04 |
> |          | Change-Param |     \-     |   49.97    |   74.96    |
> |          |     AUC      | 98.79±0.02 | 98.31±0.03 | 97.38±0.04 |
> |  PubMed  |      AP      | 98.71±0.03 | 98.15±0.06 | 96.81±0.06 |
> |          | Change-Param |     \-     |   49.84    |   74.76    |
>
> **Graph classification:**
>
> |    Dataset    |    Metric    |  Original  |    Half    |  Quarter   | 2-Original |   2-Half   | 2-Quarter  |
> | :-----------: | :----------: | :--------: | :--------: | :--------: | :--------: | :--------: | :--------: |
> |     MUTAG     |     ACC      | 87.56±0.43 | 85.36±0.42 | 84.51±0.61 | 85.02±0.92 | 84.92±1.00 | 83.64±1.13 |
> |               | Change-Param |     \-     |   72.08    |   91.53    |   64.61    |   69.99    |   72.68    |
> |  IMDB-BINARY  |     ACC      | 75.32±0.56 | 74.40±0.64 | 73.54±0.36 |     \-     | 75.34±0.54 | 75.28±0.65 |
> |               | Change-Param |     \-     |   71.11    |   90.83    |     \-     |   14.16    |   21.23    |
> |  IMDB-MULTI   |     ACC      | 52.12±0.42 | 50.71±0.55 | 48.55±0.60 | 52.05±0.53 | 52.01±0.69 | 51.91±0.72 |
> |               | Change-Param |     \-     |   73.23    |   92.42    |   37.38    |   46.73    |   51.41    |
> | REDDIT-BINARY |     ACC      | 88.24±0.12 | 85.93±0.61 | 82.47±0.98 |     \-     | 88.03±0.19 | 87.93±0.21 |
> |               | Change-Param |     \-     |   69.70    |   89.78    |     \-     |   13.21    |   19.82    |
>
> >[W\#2]  The authors point out that the proposed model is orthogonal to other fine tuning methods. A detailed discussion or more experimental results would be a plus.
>
> **Response:** Thanks for your thoughtful feedback and interest in our research.
>
> In our study, we have explored how SLIDE differs from previous fine-tuning approaches. Specifically, to demonstrate the orthogonality of our method with traditional fine-tuning approaches, we employ the following approach: we started with the classic fine-tuning method LoRA on GraphMAE, where our SLIDE model randomly prunes a subset of neurons to create frozen Slim GNNs. Subsequently, we introduce an additional LoRA module specifically designed for fine-tuning. To further validate our approach, we apply the de-correlation methods to adjust the LoRA module, aiming to reduce the correlation between embeddings within the Slim GNNs integrated with LoRA modules. The results are shown in the table, where "Slim-LoRA" denotes direct fine-tuning of Slim GNNs using LoRA and "SLIDE-LoRA" is our method.
>
> | Dataset  | Metric | Linear-probing |       LoRA        | Slim-LoRA  |    SLIDE-LoRA     |
> | :------: | :----: | :------------: | :---------------: | :--------: | :---------------: |
> |   Cora   |  ACC   |   83.96±0.12   | 84.18±0.34 | 83.62±0.29 |  **84.26±0.43**   |
> | CiteSeer |  ACC   |   73.26±0.24   | 73.27±0.36 | 72.88±0.51 |  **73.37±0.57**   |
> |  PubMed  |  ACC   |   80.62±0.17   |  **80.69±0.61**   | 80.36±0.63 | 80.63±0.65 |
>
> SLIDE's reduction of model parameters enables fine-tuning of the entire model, highlighting a distinct advantage. However, in SLIDE-LoRA which applies SLIDE based on LoRA, the parameters of Slim GNNs cannot be adjusted; only the parameters of LoRA modules can be modified. This limitation impacts SLIDE-LoRA's performance. Nevertheless, LoRA serves as a coarse method for adjusting model parameters, enabling SLIDE-LoRA to enhance model performance by reducing correlations among final representations. We achieve slightly superior results compared to using LoRA directly on Original GNNs, underscoring the efficacy of SLIDE-LoRA in enhancing model capabilities. The experiment provides empirical evidence supporting the feasibility and efficacy of our method in enhancing model performance and the orthogonality of our method with traditional fine-tuning approaches.
>
> Once again, we sincerely appreciate your valuable feedback and hope that these insights clarify the distinctiveness and effectiveness of SLIDE in the context of fine-tuning methodologies.

---

> > ### Comment · Reviewer_nfbW · 2024-08-11
> >
> > I would like to thank the author for the response, which has solved my concerns. Hence, I would like to raise my score.

---

### Official Review · Reviewer_WSVe · 2024-07-15

**Soundness:** 3
**Presentation:** 3
**Contribution:** 3
**Rating:** 5
**Confidence:** 4

**Summary:**

This paper studies the redundancy issue of GNNs that are pre-trained in the self-supervised manners. Examples of these methods include GraphMAE and GRACE. The first part of this paper shows that the numbers of parameters of these models can be reduced by half, while their performance wouldn’t change much (only drops to 96.2% of the original GNNs). This is a very  interesting finding.
The second part of the paper presents a pre-training and fine-tuning paradigm called SLIDE. It aims to fine-tune a slim GNN (a half-sampled pre-trained GNN), along with a prediction layer and weights of the nodes, so the node classification performance could be improved (comparing to other fine-tuning method).

**Strengths:**

1.	The finding presented in the first part of the paper is very interesting. The redundancy found in pre-trained weights of GNN via self-supervised learning could help simplification of GNNs and their deployment in low-resource computing environments.
2.	The empirical study of the redundancy issue in the first part is reasonable, and covers two representative graph self-supervised learning models (GraphMAE  and GRACE). Therefore, the results are meaningful and insightful.
3.	The paper overall is well written, and easy to follow.

**Weaknesses:**

1.	The  study of the redundancy issue in the first part is  solely on the node classification task. Although node classification is the most common problem in graph learning, GNN pre-trained models are also widely used for link prediction and the entire graph classification. The further study of the redundancy issue for other tasks would make this work more solid.
2.	The second part of the work (the SLIDE model) is a less valuable contribution. It is unclear about the motivation for “de-correlating the learned embeddings H in the fine-tuning phase, making models with fewer parameters more informative”. The “Slim GNN” preserved most of the useful weights already. Encouraging the independence of the node embeddings or not doesn’t change much the performance in downstream tasks. In Fig. 5, models without “de-correlation” only have a slight performance decrease. However, including “de-correlation” can level up the computational complexity.
3.	The comparison in Fig.6 with “full fine-tune” is unfair. Why not including the  “linear probing” for comparison?  fine-tuning only the additional classifier and freezing the original GNNs). The number of parameters in “linear probing” is much less, while the performance of “linear probing” is not that bad (as shown in Table 3-5).

**Questions:**

While I really like the first part of the work, I have the following questions:
1.	What datasets were used for pre-training these GNNs before getting the slim GNNS?
2.	Any possibility to provide theoretical analysis for the redundancy issue?
3.	Did authors run statistical tests for comparing the performance of LP, FT and SLIDE in Table 3-5?  The performance drop indicated in red color is often smaller than the std. So it is unknown if the performance difference is statistically significant.

---

> ### Author Rebuttal · Authors · 2024-08-06
>
> We sincerely thank you for valuable opinions and concerns about our work. It is our obligation to describe more details and give more explanations. We really hope that these further efforts can alleviate your confusion.
>
> >[W\#1] About the issue of redundancy
>
> **Response:** To demonstrate that redundancy exists across a broader spectrum of graph learning tasks, we conduct experiments on GraphMAE, effective for graph classification, and MaskGAE which excels in link prediction. The results indicate that model redundancy exists across a wide range of tasks.
>
> Link prediction:
>
> |Dataset|Metric|Original|Half|Quarter|
> |:------:|:----------:|:--------:|:--------:|:--------:|
> ||AUC|96.7|96.3|93.8|
> |Cora|AP|96.2|96.2|94.0|
> ||Change-Param|\-|49.9|74.9|
> ||AUC|97.8|97.1|95.5|
> |CiteSeer|AP|98.1|97.4|96.3|
> ||Change-Param|\-|50.0|75.0|
> ||AUC|98.8|98.3|97.4|
> |PubMed|AP|98.7|98.2|96.8|
> ||Change-Param|\-|49.8|74.8|
>
> Graph classification:
>
> |Dataset|Metric|Original|Half|Quarter|2-Original|2-Half|2-Quarter|
> |:-----------:|:----------:|:--------:|:--------:|:--------:|:--------:|:--------:|:--------:|
> |MUTAG|ACC|87.6|85.4|84.5|85.0|84.9|83.6|
> ||Change-Param|\-|72.1|91.5|64.6|70.0|72.7|
> |IMDB-B|ACC|75.3|74.4|73.5|\-|75.3|75.3|
> ||Change-Param|\-|71.1|90.8|\-|14.2|21.2|
> |IMDB-M|ACC|52.1|50.7|48.6|52.1|52.0|51.9|
> ||Change-Param|\-|73.2|92.4|37.4|46.7|51.4|
> |REDDIT-B|ACC|88.2|85.9|82.5|\-|88.0|87.9|
> ||Change-Param|\-|69.7|89.8|\-|13.2|19.8|
>
> >[W\#2] About SLIDE
>
> **Response:** Thanks. We appreciate the opportunity to clarify the contributions and the motivations behind our approach.
>
> Our research has demonstrated the existence of model redundancy in graph self-supervised learning models. Even though we remove a portion of parameters, model redundancy may still occur during fine-tuning. Therefore, based on our findings, we believe that it is necessary to introduce a de-correlation module during fine-tuning. Our experiments have also shown that integrating such a module improves the performance. We observe that models without de-correlation show a noticeable decrease in performance compared to models with de-correlation by conducting statistical significance tests. Specifically, we present the p-values from experiments conducted across 6 datasets using MaskGAE. As can be seen, p-values are less than 0.01 on most of them. These results underscore the statistical significance of the performance differences between models with and without de-correlation.
>
> |Cora|CiteSeer|PubMed|Photo|Computers|arxiv|
> |:------:|:------:|:------:|:----------:|:--------------:|:--------:|
> |*0.00026|*0.013|*0.0032|*0.0013|*0.027|*0.0095|
>
> Regarding computational complexity, the de-correlation method we introduced maintains linear complexity relative to the number of nodes. This ensures that the additional computational cost is minimal compared to fine-tuning Slim GNNs without de-correlation, preserving the efficiency of the models.
>
> >[W\#3] About comparison
>
> **Response:** Thanks.
>
> The motivation behind Fig. 6 is to show that despite reducing the parameters, SLIDE maintains comparable performance. We show that our approach effectively reduces the parameter tuning required compared to full fine-tuning of Original GNNs.
>
> Actually, in Section 2 we have already compared the performance and parameter reduction achieved by "linear probing" between Original GNNs and Slim GNNs. Only classifiers are fine-tuned while GNNs are frozen. This results of the comparisons are explicitly detailed in Table 1-2, which also implicitly indicate the proportion of parameters attributed to linear layers (50% and 25%). Furthermore, our experimental results indicate that despite significant parameter reduction, the performance of Slim GNNs does not exhibit a noticeable degradation.
>
> >[Q\#1] About datasets
>
> **Response:** As mentioned in Section 2, we pretrain Original GNNs on six datasets (Cora, CiteSeer, PubMed, Amazon-Photo, Amazon-Computers and Ogbn-arxiv), and obtain several kinds of Slim GNNs.
>
> >[Q\#2] About theoretical analysis
>
> **Response:** Thanks.
>
> As highlighted in the conclusion, we agree with you and recognize the importance of theoretical analysis in enhancing our understanding. Theoretical analysis of model redundancy represents a significant avenue for future research. Enhancing the theoretical foundations of model redundancy will be crucial for advancing our methodologies.
>
> We could potentially explore theoretical models from several angles. For instance, we could theoretically analyze model redundancy from the perspectives of feature decorrelation. Investigating causal mechanisms within GNN architectures could also offer formal insights into how redundant features affect model predictions. Additionally, developing mathematical formulations to quantify redundant features could offer intuitive insights and further optimize GNN efficiency.
>
> We appreciate your insightful question and recognize theoretical analysis as a critical next step in advancing our understanding and methodologies for addressing redundancy within graph self-supervised learning models.
>
> >[Q\#3] About statistical tests
>
> **Response:** We have performed significance tests and calculated p-values to evaluate the performance differences between our method and previous approaches. The p-values for most of comparisons are less than 0.05, indicating that the performance differences are statistically significant. As an example, we present significance test results for MaskGAE in the table below.
>
> ||Cora|CiteSeer|PubMed|Photo|Computers|Arxiv|
> |----|:------:|:------:|:------:|:----------:|:--------------:|:-------:|
> |LP|*0.00003|*0.00043|0.42|*0.00002|*0.0000003|*0.011|
> |FT|*0.0035|*0.0077|*0.035|*0.046|*0.0086|0.91|
>
> Observing the results, out of 12 comparisons, 10 pass the significance test with p-values less than 0.05. One exception is observed in the comparison between SLIDE and FT on Ogbn-arxiv, where SLIDE achieves comparable performance to FT despite using parameters that are 67% fewer.

---

### Decision · Program_Chairs · 2024-09-25

**Decision:**

Accept (poster)

**Comment:**

All four reviewers agreed this paper should be accepted: it identifies an important problem, the paper is clearly written, and the proposed method is simple and effective. The paper will likely provide useful insights for future work in graph self-supervised learning methods. This is a clear accept. Authors: you've already responded to the detailed reviewer feedback, please make any changes to your manuscript to further improve the paper based on the feedback from reviewers. The paper will make a great contribution to the conference!